# Deep End-to-end Unsupervised Anomaly Detection

## Abstract

This paper proposes a novel method to detect anomalies in large datasets under a fully unsupervised setting. The key idea behind our algorithm is to learn the representation underlying normal data. To this end, we leverage the latest clustering technique suitable for handling high dimensional data. This hypothesis provides a reliable starting point for normal data selection. We train an autoencoder from the normal data subset, and iterate between hypothesizing normal candidate subset based on clustering and representation learning. The reconstruction error from the learned autoencoder serves as a scoring function to assess the normality of the data. Experimental results on several public benchmark datasets show that the proposed method outperforms state-of-the-art unsupervised techniques and is comparable to semi-supervised techniques in most cases.

## 1 Introduction

Anomaly detection refers to the identification of patterns that do not conform to expected normal behavior (Chandola et al. (2009)). It is a critical task in diverse application domains such as fraud detection (Phua et al. (2010)), intrusion detection (Lazarevic et al. (2003)) and surveillance video profiling (Xiang & Gong (2008); Saligrama et al. (2010)). While the concept of an anomaly is intuitively easy for humans to understand, it is hard to define mathematically. Fundamentally, an anomaly is something with insufficient similarity to the rest of the data. This similarity can be computed on the basis of some feature difference. However, what makes an ideal feature representation for the data depends on what constitutes an anomaly. This forces anomaly detection into a chicken-or-egg problem in which there are a pair of problems, neither of which can be solved before the other.

To date, a number of works have attempted this problem by training an autoencoder to create low-dimensional representations for anomaly detection (Chalapathy et al. (2018); Zhai et al. (2016); Zong et al. (2018)). The anomalies are rejected and the autoencoder retrained (Mishne & Cohen (2017); Wang et al. (2017)). While this gives reasonable results, it is fundamentally dependent on how well the first iteration solves the problem.

We propose a solution in which anomalies can be defined using approximately correct features. This is achieved through an observation. Given a feature, anomalies approximately correspond to instances of high variance distributions. Such instances can be identified using a distribution-clustering (Lin et al. (2018)) framework. This hypothesis provides a reliable starting point for normal data selection. We train an autoencoder from the normal data subset, and iterate between hypothesizing normal candidate subset and representation learning. The reconstruction error serves as a scoring function to assess the normality of the data. The proposed framework does not rely on any training labels. Instead, it iteratively distills out anomalous data and improves the learned representation of normal data by incorporating clustering techniques into the process.

We extensively assess the broad applicability of the proposed model on network intrusion, image and video data. Empirical results show that the proposed method outperforms the existing state-of-art approaches in terms of both accuracy and robustness to the percentage of anomalous data.

## 2 Related Works

Existing anomaly detection methods can be grouped into three categories.

**Reconstruction-based method** These methods assume that anomalies are incompressible and thus cannot be effectively reconstructed from low-dimensional projections. Classical methods like Principle Component Analysis(PCA) (I.T.Jolliffe (1986)) and Robust-PCA (Candès et al. (2011)) are motivated by this assumption. In recent works, different forms of deep autoencoder are proposed to analyze the reconstruction error. Xia et al. (Xia et al. (2015)) show that by introducing a regularizing term to a convolutional autoencoder, the anomalies tend to produce a bigger reconstruction error. Variational Autoencoder (VAE) (An & Cho (2015)) and Generative Adversarial Networks (GANs) (Schlegl et al. (2019)) have also been introduced to perform reconstruction-based anomaly detection. These methods demonstrate promising results when the anomaly ratio is fairly low. Although the reconstruction of anomalous samples, based on a reconstruction scheme optimized for normal data, tends to generate a higher error, a significant amount of anomalous samples could mislead the autoencoders to learn the correlations in the anomalous data instead.

**Density estimation and clustering** Motivated by the assumption that anomalies occur less frequently, these algorithms treat anomalies as low-density regions in some feature space. Clustering analysis, such as Robust-KDE (JooSeuk Kim & Scott (2008)), is often used for density estimation and anomaly detection. Unfortunately, due to the curse of dimensionality, these methods are less applicable to analysing high-dimensional data, where density estimation is a challenge in itself.

A two-step approach is normally adopted to counter this issue, where dimensionality reduction is conducted first, followed by clustering analysis as a separate step. One drawback of this approach is that dimensionality reduction is trained without the guidance from the subsequent clustering analysis; hence the key information for clustering analysis could be lost during dimensionality reduction. Recent works jointly learn dimensionality reduction and clustering components based on deep autoencoder (Zhai et al. (2016); Zong et al. (2018)). Notably, DAGMM (Zong et al. (2018)) utilizes an autoencoder to generate a low-dimensional representation and its reconstruction error, which is further fed into an estimation network based on Gaussian Mixture Model(GMM). However, as its autoencoder was trained on the whole dataset, it is vulnerable to a high percentage of anomalous samples and may learn wrong correlations. In contrast, our proposed method addresses this issue by first finding a normal candidate subset to train an autoencoder and then iterating between representation learning and refinement of the normal candidate.

**One-class classification** One-class SVM (Erfani et al.; Chalapathy et al. (2018)) is also widely used. Under this framework, a discriminative boundary surrounding the normal instances is learned by algorithms. However, when dimensionality goes higher, such techniques often suffer from suboptimal performance due to the curse of dimensionality. OCNN (Chalapathy et al. (2018)) attempted to circumvent this problem by using an autoencoder for dimensionality reduction. However, OCNN requires training data with relatively low anomaly ratio, in order to obtain an optimized NN model to differentiate anomalies from single-class normal data.

## 3 PROBLEM FORMULATION

Let $\mathbf{X} = \{\mathbf{x}_i\}, i = 1, \ldots, N, \mathbf{x} \in \mathbb{R}^k$ be the set of input data points that contains a certain percentage of anomaly. The goal of anomaly detection is to learn a scoring function $h(\mathbf{x})$, $h : \mathbb{R}^k \mapsto \mathbb{R}$, to classify samples $\mathbf{x}_i$ based on some threshold $\lambda$:

$$y_i = \begin{cases} 0, & \text{if } h(\mathbf{x}_i) < \lambda \\ 1, & \text{if } h(\mathbf{x}_i) \geq \lambda \end{cases} \tag{1}$$

where $y_i$ are the labels. $y_i = 0$ indicates $\mathbf{x}_i$ is normal and $y_i = 1$ indicates anomalous.

An overview of the proposed end-to-end anomaly detection system is presented in Fig. 1. The major component of this system is an autoencoder that learns a low-dimensional representation of the input data that are often of high dimensions, to enable simplified modeling of the underlying distribution of the data. Under a fully unsupervised setting, the only information we are given is the set of input data $\mathbf{X}$, without any label information. As an initialization, we leverage the latest clustering technique for high-dimensional data (Lin et al. (2018)) to provide soft supervisory signals.

Since our input data is unlabelled, we derive a "training" set $\mathbf{S}_{train}$, where $\mathbf{S}_{train} \subset \mathbf{X}$ based on the following:

$$\mathbf{S}_{train} = \mathcal{C}(\mathbf{X}, p_0) \tag{2}$$

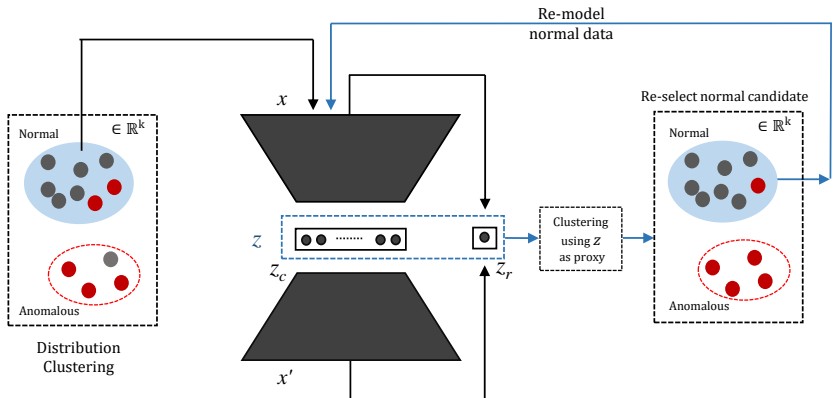

Figure 1: Flow-chart of the proposed end-to-end anomaly detection system.

where $\mathcal{C}$ represents a selection process based on clustering output, and $p_0$ represents the percentage of anomaly, it controls which are the clusters to be accepted into the "training" set. In our experiments, we compute the threshold as the $(100 - p_0)^{th}$ percentile of cluster variance, and accept clusters with variance smaller than this threshold. The assumption here is that clusters with large variance are likely to contain anomalous members.

### 3.1 SCORING FUNCTION LEARNING

The autoencoder network provides two sources of features: (1) a low-dimensional representation of the original input data; and (2) the reconstruction error by comparing the input with its decoded counter-part. Using the training set $\mathbf{S}_{train} = \{\mathbf{s}_1, \mathbf{s}_2, \cdots, \mathbf{s}_M\}$, the autoencoder learns the encoding function $f_{en}$:

$$\mathbf{z}_c = f_{en}(\mathbf{s}; \Theta_{en}), \quad \forall \mathbf{s} \in \mathbf{S}_{train}, \quad \mathbf{z}_c \in \mathbb{R}^{k_{bn}} \tag{3}$$

where $\Theta_{en}$ are the learned parameters for the encoder. $\mathbf{z}_c$ are known as the bottle-neck features of dimension $k_{bn}$.

Similarly, for the decoding part, we have:

$$\mathbf{x}' = f_{de}(\mathbf{z}_c; \Theta_{de}), \tag{4}$$

where $\Theta_{de}$ are the learned parameters for decoding. $\mathbf{x}'$ are the reconstructed features.

Upon training, we have a learned autoencoder with optimized parameters $\{\Theta_{en}, \Theta_{de}\}$. We run the entire input set $\mathbf{X}$ forward through the encoder network and produce a set of new features $\mathbf{Z} = \{\mathbf{z}_1, \mathbf{z}_2, \cdots, \mathbf{z}_N\}$, by concatenating the bottle-neck feature with the reconstruction error:

$$\mathbf{z} = [\mathbf{z}_c; \mathbf{z}_r], \quad \mathbf{z} \in \mathbb{R}^{k_{bn}+1}, \tag{5}$$

where the reconstruction error $\mathbf{z}_r$ is measured in terms of cosine similarity between $\mathbf{x}$ and its decoded counter-part:

$$\mathbf{z}_r = d(\mathbf{x}, \mathbf{x}') = \cos^{-1}\left(\frac{\mathbf{x}^T \mathbf{x}'}{\| \mathbf{x} \| \| \mathbf{x}' \|}\right) \tag{6}$$

Note that $\mathbf{Z}$ is now of a much lower dimension than the input data $\mathbf{X}$. Hence, traditional clustering techniques such as Gaussian Mixture Model would suffice for subsequent training set selection. To ensure the initial training set can capture most of the normal samples, we adopt a more conservative cluster variance threshold.

With the new encoding scheme, the entire input set $\mathbf{X}$ is now represented as $\mathbf{Z}$. We can "re-label" the training set $\mathbf{X}$ by using $\mathbf{Z}$ as a proxy. Similar to the initial training set selection, we select members that belong to low-variance clusters in $\mathbf{Z}$. The process of *Training set selection → Autoencoder training → New feature computation* is performed iteratively. The training set is updated as follows:

$$\mathbf{Z}_{train}^{t+1} = \mathcal{C}(\mathbf{Z}^t, p), \tag{7}$$

$$\mathbf{S}_{train}^{t+1} = \{\mathbf{x}_j : \forall \mathbf{z}_j \in \mathbf{Z}_{train}^{t+1}\}, \tag{8}$$

where the superscript $t$ here refers to the $t^{th}$ iteration.

Finally, the training process terminates when there is no change in the set of selected normal samples between two successive iterations. After the last iteration, $t = t_F$, we obtain the autoencoder parameters $\{\Theta_{en}^{t_F}, \Theta_{de}^{t_F}\}$, and use it to construct the scoring function:

$$h(\mathbf{x}) = d(\mathbf{x}, \mathbf{x}') = d(\mathbf{x}, f_{de}(f_{en}(\mathbf{x}; \Theta_{en}^{t_F}); \ \Theta_{de}^{t_F})), \tag{9}$$

where $\mathbf{x}'$ is the result of going through the encoding-decoding process according to the trained autoencoder.

## 3.2 ALGORITHM

The proposed framework is summarized in Algorithm 1. We obtain an initial split of the data into normal and abnormal subsets through clustering (i.e. GMM for KDDCUP data and Distribution Clustering (Lin et al. (2018)) for image and video data). Selected outputs for CIFAR-10 and MNIST datasets using Distribution Clustering are presented in Fig. 4 in the appendix. We observe that as cluster variance increases, the samples' appearance become more anomalous.

---

**Algorithm 1** Deep end-to-end Unsupervised Anomaly Detection

---

**Input:** $\mathbf{X} = \{\mathbf{x}_i\}$, $i = 1, 2..., N$: set of normal and anomalous input examples.
   $r$: number of epochs required for re-evaluation of the membership of the entire input set $\mathbf{X}$.
   $p_0$ and $p$: thresholds for initial and subsequent training set selection, respectively
**Output:** Reconstruction-based anomaly score function $h(\mathbf{x})$ and trained autoencoder $\{\Theta_{en}^{t_F}, \Theta_{de}^{t_F}\}$,
1: **procedure** GET_DECISION_SCORE($\mathbf{X}, r, p, f_{en}, f_{de}$)
2:  $\mathbf{S}_{train} \leftarrow \mathcal{C}(\mathbf{X}, p_0)$    ▷ Run clustering, select instances from low-variance clusters
3:  $\mathbf{L} = \{k : \forall \mathbf{x}_k \in \mathbf{S}_{train}\}$    ▷ $\mathbf{L}$ is the set of indices of selected normal training samples
4:  $\mathbf{L}^{old} := \emptyset$
5:  **while setdiff**($\mathbf{L}^{old}, \mathbf{L}$) $\neq \emptyset$ **do**
6:    **for** each epoch **do**
7:      **if** $((CurrentEpoch + 1) \bmod \mathbf{r}) == 0$ **then** ▷ Re-evaluate normality every r epochs
8:        $\mathbf{Z}_c \leftarrow f_{en}(\mathbf{X}, \Theta_{en})$               ▷ Bottle-neck features
9:        $\mathbf{X}' \leftarrow f_{de}(\mathbf{Z}_c, \Theta_{de})$
10:       $\mathbf{Z}_r \leftarrow d(\mathbf{X}, \mathbf{X}')$                  ▷ Reconstruction error
11:       $\mathbf{Z} \leftarrow [\mathbf{Z}_c; \mathbf{Z}_r]$
12:       $\mathbf{S}_{train} \leftarrow \mathcal{C}(\mathbf{Z}, p)$        ▷ Get new training set according to threshold $p$
13:       $\mathbf{L}^{old} := \mathbf{L}$
14:       $\mathbf{L} \leftarrow \mathbf{S}_{train}$              ▷ Update set of indices for training samples
15:     **else**
16:       Train $f_{en}, f_{de}$ on $\mathbf{S}_{train}$ to obtain $\{\Theta_{en}, \Theta_{de}\}$
17:    **end for**
18:   **end while**
19:   $\Theta_{en}^{t_F} = \Theta_{en}, \quad \Theta_{de}^{t_F} = \Theta_{de}$
20:   **Output** $h(\mathbf{x})$ according to finalized autoencoder $\{\Theta_{en}^{t_F}, \Theta_{de}^{t_F}\}$ base on Eq. (9)
21: **end procedure**

---

**Convergence**   Assuming a $p\%$ anomaly ratio, our algorithm starts with a tight cut-off, accepting clusters with variances below $(100 - p_0)^{th}$ percentile as an initial training set, where $p_0 > p$. This ensures the initial training set is as pure as possible. Our assumption is that given partial normal data, the autoencoder would be able to learn a representation and generalize well on the "unseen" normal data that was discarded, and progressively recover them as iterations go on. Empirically, we plotted the AUROC, AUPRC and F-score for 20 iterations for the KDDCUP experiment, presented in Fig. 5 of the appendix. It demonstrates the convergence as iteration progresses. The same behavior was observed throughout our experiments on other datasets.

## 4 EXPERIMENTS

### 4.1 BASELINE METHODS

On the topic of anomaly detection, there exist different terminologies concerning the nature of supervision: (a) Algorithm uses label information of the normal class for training (label information could be used in part, or all of the stages of an algorithm); (b) No training labels are given, algorithm treats the entire dataset with both normal and anomalous classes as input. For the purpose of this paper, we term type (a) semi-supervised and type (b) unsupervised. We evaluate our proposed algorithm against the following state-of-the-art methods:

**OC-NN** *(semi-supervised)* One-class neural networks (OC-NN) (Chalapathy et al. (2018)) contains 2 major components: a deep autoencoder and a feed-forward convolutional network. The deep encoder is trained on normal data for representation learning. The trained encoder, with its parameters frozen, is subsequently used as the input layers of a feed-forward network with 1 extra hidden layer. Variants of OC-NN employ different activation functions (i.e. linear, sigmoid, relu) in the hidden layer. We report the best score attained among all possible activation functions in our experiments.

**OC-SVM** *(unsupervised)* One-class support vector machine (OC-SVM) (Erfani et al.) is a kernel-based method for anomaly detection. The algorithm searches for best-performing hyper-parameters $\gamma$ (kernel coefficient) and $\nu$ (upper bound of the fraction of training errors and lower bound of the fraction of support vectors) to obtain the optimal AUROC (Buitinck et al. (2013)).

**DAGMM** *(unsupervised)* Deep autoencoding Gaussian mixture model (DAGMM) (Zong et al. (2018)), comprised of one compression net and one estimation net, is a method based on representation learning. The compression network provides low-dimensional representations of input samples and the reconstruction error features. They are fed into the estimation network, which functions as a Gaussian Mixture Model, to predict the mixture membership for each sample. We modify the original DAGMM algorithm by adding a small value to the diagonal elements of the covariance matrix. The model achieves better results than the reported score from the original work.

**Deep anomaly detection using geometric transformations** *(semi-supervised)* This method (Golan & El-Yaniv (2018)) employs a deep neural model to identify out-of-distribution samples of image data, given only the examples from the normal class. A series of geometric transformations are applied to the normal class to create a multi-class dataset. A deep neural net, trained using this dataset, is then employed to discriminate the transformations applied. Subsequently, given an unseen instance, the model applies each transformation on it and assigns membership scores. The final normality score is determined based on the combined log-likelihood of softmax response vectors.

### 4.2 DATASETS

We employ five benchmark datasets, namely, KDDCUP, MNIST, CIFAR-10, CatVsDog and UCF-Crime, to evaluate our proposed method, together with other methods described above.

• **KDDCUP:** The KDDCUP network intrusion dataset (Lichman et al. (2013)) contains samples of 41 dimensions. Similar to (Zong et al. (2018)), categorical features are prepared by applying one-hot encoding. 20% of the "normal" samples form the minority group, while the rest 80% are treated as "attackers". As "normal" samples are the minorities, they are treated as anomalies

• **MNIST:** The MNIST dataset (LeCun & Cortes (2010)) consists of 60,000 gray-scale $28 \times 28$ images of handwritten digits from 0 to 9. We formulate an anomaly detection task as per described in (Chalapathy et al. (2018)) and (Zhou & Paffenroth (2017)), where 4,859 images of digit 4 are randomly sampled as normal instances and 265 images are evenly sampled from all other categories as anomalies.

• **CatVsDog:** The CatVsDog dataset consists of dogs and cats images of varying sizes, which are extracted from the ASIRRA dataset (Elson et al. (2007)) following the settings specified in (Golan & El-Yaniv (2018)). 12,500 images of dogs and 2,500 images of cats are sampled to form an anomaly detection task. The cat images are treated as anomalies.

• **UCF-Crime:** The UCF-Crime dataset (Sultani et al. (2018)) contains 1,900 long and untrimmed videos captured from CCTV cameras. It covers 13 real-world anomalies, including incidents like fighting, burglary, abuse and etc. In both of the training and testing sets, videos are of different length and anomalies happens at various temporal locations. Some of the videos may have multiple anomalies. In the experiment, we randomly select reasonably long videos with at least 1000 frames and anomaly ratio less than 0.35 across 4 categories.

Table 1: Summary statistics of datasets.

| Dataset | Normal Class | Input Dimension | # Instances | Anomaly Ratio (%) |
|---|---|---|---|---|
| KDDcup | attack | 1×120 | 494,021 | 20 |
| MNIST | digit 4 | 28×28 | 5,105 | 3 |
| CIFAR-10 | airplane (category 0) | 32×32 | 5,500 | 8 |
| CatVsDog | dog | 128×128 | 15,000 | 17 |
| UCF-Crime | non-crime scenes | varying | dep. on video | < 35 |

## 4.3 EVALUATIONS

We adopt Area Under the curve of the Receiver Operating Characteristic (AUROC) as the main evaluation metric to measure the discrimination power of different models. AUROC is a standard method to assess the effectiveness of a classifier (Fawcett (2006)). It can be interpreted as the probability that an anomalous instance is assigned to a higher anomaly score than a normal instance (Davis & Goadrich (2006)). In this section, we compare the performance of our method against other baseline methods.

**KDDCUP: Network Intrusion Data** In this experiment, we divide the KDDCUP dataset following the setting in (Zong et al. (2018)). $50\%$ of the data is reserved for testing by random sampling. From the remaining $50\%$ of the data reserved for training, we take all samples from the normal class and mix them with different percentages of samples from the anomaly class to form the training set. Parameters for this experiment (see Algorithm 1) are set to: $p_0 = 35\%$, $p = 30\%$, $r = 10$.

Table 2 and 3 reports the AUROC and AUPRC of OC-SVM, DAGMM and our model on the KDDCUP dataset after 200 epochs, with anomaly ratio in training set being $5\%$, $10\%$ and $20\%$, respectively. It can be observed that the increase in ratio of anomalous data undermines the detection performance of OC-SVM and DAGMM more severely, while our method remains robust to such changes.

Table 2: AUROC of different models with different anomaly ratio based on KDDCUP dataset (in %). Our proposed method is much more immune to increase in anomaly ratio.

| Anomaly Ratio (%) | OC-SVM | DAGMM | Ours |
|---|---|---|---|
| 5 | $96.8 \pm 0.5$ | $96.6 \pm 1.1$ | $\mathbf{98.2 \pm 1.0}$ |
| 10 | $89.7 \pm 0.1$ | $88.6 \pm 2.0$ | $\mathbf{98.4 \pm 0.8}$ |
| 20 | $61.6 \pm 0.1$ | $79.5 \pm 2.0$ | $\mathbf{93.5 \pm 1.1}$ |

Table 3: AUPRC of different models with different anomaly ratio based on KDDCUP dataset (in %). Our proposed method is much more immune to increase in anomaly ratio.

| Anomaly Ratio (%) | OC-SVM | DAGMM | Ours |
|---|---|---|---|
| 5 | $77.8 \pm 0.1$ | $75.4 \pm 0.7$ | $\mathbf{94.5 \pm 0.1}$ |
| 10 | $68.1 \pm 0.1$ | $53.4 \pm 2.7$ | $\mathbf{93.9 \pm 0.1}$ |
| 20 | $45.4 \pm 0.0$ | $40.7 \pm 2.8$ | $\mathbf{90.3 \pm 0.5}$ |

Figure 2 shows the Receiver Operating Characteristic (ROC) curves of different models when the anomaly ratio of the training data is $20\%$. In our unsupervised setting where no prior knowledge of normal class is known, our method is clearly more robust to contaminated training data.

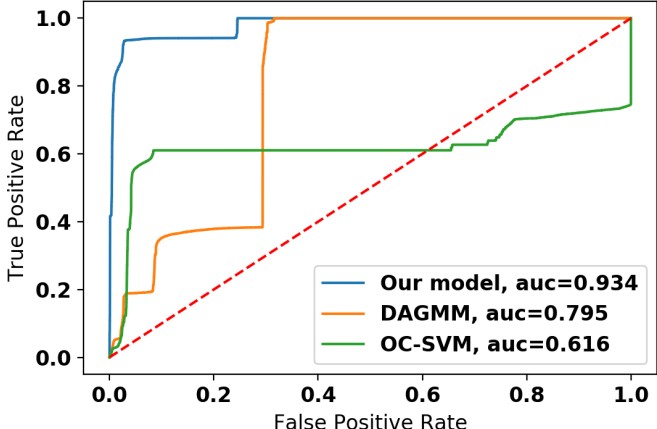

Figure 2: ROC comparison of our proposed model, DAGMM, and OC-SVM. Results are obtained based on the KDDCUP dataset, with 20% anomaly ratio.

Table 4: AUROC in %. Highest score among all methods and highest score among all unsupervised methods are highlighted. On complex datasets such as CIFAR-10 and CatVsDog, our proposed method has higher performance gain among all unsupervised methods. The last column presents results obtained using distribution clustering alone.

| Dataset | OC-NN (semi-supervised) | OC-SVM (unsupervised) | DAGMM (unsupervised) | Geom. Transform. (semi-supervised) | Ours (unsupervised) | Distrib. Clust. |
|---|---|---|---|---|---|---|
| MNIST | 70.0 | **90.2** | 50.3 | **98.2** | $70.5 \pm 2.1$ $82.4 \pm 1.8$ (raw) | 63.3 |
| CIFAR-10 | 63.8 | 69.7 | 49.0 | 73.3 | **73**.6 ± **0.6** | 48.7 |
| CatVsDog | 50.8 | 56.2 | 43.4 | **88.3** | **74**.0 ± **2.5** | 56.1 |

**Image Data**  In table 4, we compare the AUROC scores obtained from OC-NN, OC-SVM, DAGMM, Geometric Transformation and our model, based on multiple image datasets. It should be noted that Geometric Transformation approach trains on data from the normal class only (hence classified as semi-supervised). Our method, on the other hand, does not require label information.

Results in Table 4 demonstrate an outstanding performance of our method over other unsupervised approaches. In addition, on CIFAR-10, the performance of our proposed algorithm is comparable to that of Geometric Transformation, a semi-supervised method. In the last column of Table 4, we report results obtained using distribution clustering alone. The combination of distribution clustering and autoencoder significantly improves the discrimination against anomalies. Details of the parameters used in the experiments are reported in the appendix.

We make several key observations based on the attained results in Table 4. Unlike all other methods that tend to perform better on simpler datasets (e.g. MNIST), the advantage of our method becomes more evident on datasets with higher complexity. Notably, our method outperforms other unsupervised approaches on CIFAR-10 and CatVsDog. The shortfall of our method on MNIST dataset could be due to the adoption of NetVLad feature extractor (4096-d feature vectors), which may not be an ideal choice of feature representation since the images are pre-aligned and hence of low dimensionality. We repeat the same experiment using raw image features from MNIST. It shows improved AUROC score, suggesting that raw feature is a better representation.

We also observe that while our method is able to produce results comparable to semi-supervised approaches, the gap is wider on CatVsDog dataset as compared to CIFAR-10. We attribute this to the high noise level in the CatVsDog dataset. For example, some images consist of both dog and cats. Moreover, training on normal data (with augmentation through geometric transformation) gives

Geometric Transform a natural advantage. According to (Elson et al. (2007)), ASIRRA dataset, from which the CatVsDog is extracted, is deemed extremely challenging for computers. Sample images of the dataset are presented in the appendix.

**Video Data** We apply the proposed approach on UCF-Crime dataset (Sultani et al. (2018)), with features extracted using C3D (Tran et al. (2015)) descriptor. In default C3D settings, every 16 frames are aggregated to generate 1 feature vector. As our method is unsupervised, it needs a sufficient amount of feature vectors for training. We select videos with at least 65 vectors (1040 frames).

In (Sultani et al. (2018)), although the AUROC score of each video category is not reported, the AUROC averaged across the entire testing set of UCF-Crime dataset is $75.4\%$. It is achieved by adopting a semi-supervised method of multiple instance learning (MIL). The AUROC scores on 4 randomly selected video categories, using our method and DAGMM, are reported in Table 7. The parameters used in training, as well as individual video-level scores are detailed in the appendix.

A good correspondence between the ground truth and our anomaly scores can be observed in Figure 3, where frames with anomalous events under the orange lines are assigned to higher anomaly scores. It can be observed that our method significantly out-performs DAGMM. Moreover, our method is able to produce AUROC higher than $75.4\%$ for some action categories. Despite under a fully unsupervised setting, our method is as effective in detecting anomalies from video data as its semi-supervised competitor, demonstrating its strength in handling complex and high-dimensional data.

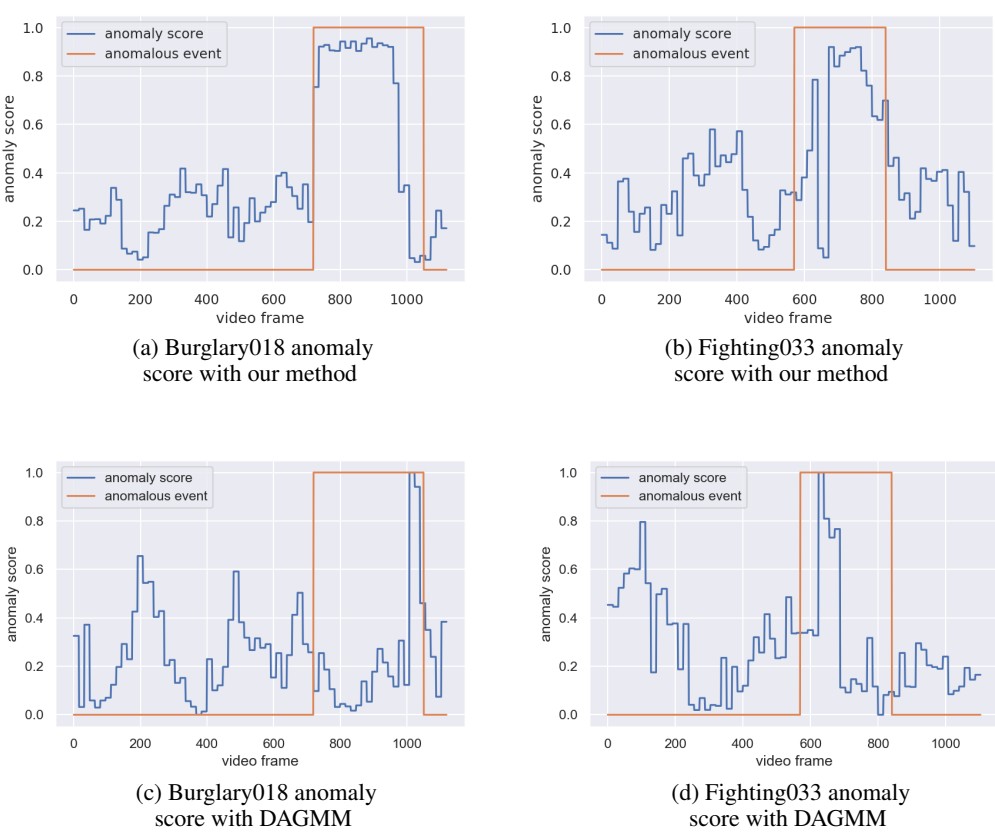

(a) Burglary018 anomaly
score with our method

(b) Fighting033 anomaly
score with our method

(c) Burglary018 anomaly
score with DAGMM

(d) Fighting033 anomaly
score with DAGMM

Figure 3: Anomaly scores (normalized) plotted against ground truth (flagged by orange lines). Compared to DAGMM, our method shows much better correspondence to the ground truth.

**Run Time** Excluding feature extraction and clustering process, on a single NVIDIA Tesla P100 GPU, our method takes 4 minutes 20 seconds on average to complete the CIFAR-10 experiment described above (consisting of 5,500 instances). This timing is averaged over 5 runs.

### 4.4 ABLATION STUDY

**Initialization**    To examine the effect of adopting distribution clustering as the initialization method for high-dimensional data, a variety of other mainstream clustering methods, including K-means, HDBSCAN (McInnes et al. (2017)) and Gaussian Mixture Model (GMM) are used to replace the distribution clustering component in the initial normal subset selection. We compare results on the CIFAR-10 task.

For K-means and GMM, prior information on the number of clusters/components is set to 20, which is consistent with the setting of the GMM employed in the proposed model. For HDBSCAN, the minimum size of a cluster is set to 5, that follows the setting as distribution clustering. Table 6 reports the AUROC scores obtained from CIFAR-10 anomaly detection task with different clustering techniques. The results demonstrate that using distribution clustering initialisation provides better supervisory signals and leads to favourable performance.

Table 5: AUROC in % of detecting crime scene as anomalies in surveillance videos.

| Crime scene | # Video selected | Ours | DAGMM |
|---|---|---|---|
| Arrest | 2 | 70.6 | 52.2 |
| Arson | 3 | 67.8 | 60.1 |
| Burglary | 4 | 79.2 | 67.4 |
| Fighting | 3 | 77.1 | 57.0 |

Table 6: Comparison on initialisation method on CIFAR-10.

| Clustering method | AUROC |
|---|---|
| K-means | 60.3 |
| HDBSCAN | 61.2 |
| GMM | 50.0 |
| Distribution Clustering | **73.6** |

To further understand the effectiveness of distribution clustering, we tabulated in the AUROC achieved using distribution clustering alone, for the experiments on image data (refer to right-most column of Table 4). Surprisingly, this result is even better than those of DAGMM and OC-NN on the challenging CatVsDog dataset.

## 5    DISCUSSION AND CONCLUSION

This paper presents an end-to-end method for anomaly detection under a fully unsupervised setting. The key insight of our algorithm is to model normal data. We leverage distribution clustering technique to make an educated guess on the normal data subset. By incorporating clustering to provide supervisory signals, we iterate between hypothesizing normal candidate subset and representation learning. This framework iteratively distills out anomalous data and improves the learned representation of normal data. Extensive experiments on benchmark datasets demonstrate our proposed method outperforms existing unsupervised approaches and is comparable to semi-supervised solutions in most cases.

**Limitations and future work**: Using only an autoencoder, our current method may be insufficient to handle highly complex patterns and hence falls short on difficult dataset such as CatVsDog. For future work, we seek to explore more sophisticated generative frameworks for representation learning.

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

# A    INITIALIZATION

Selected outputs for CIFAR-10 and MNIST datasets using Distribution Clustering are shown in Fig. 4. We observe that as cluster variance increases, the samples' appearance become more anomalous.

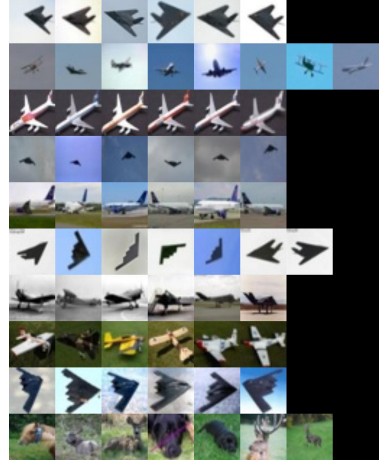 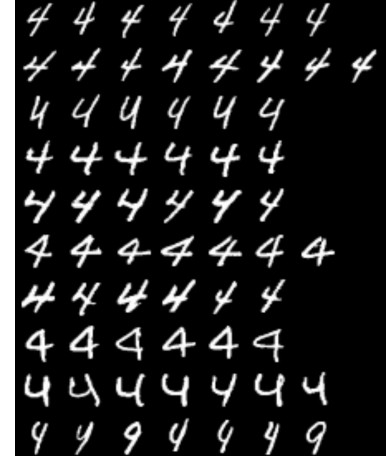

(a) Clustering result of CIFAR-10 ("airplanes" class forms the normal group) : cluster 5, 10, 15, 20, 25, 30, 35, 40, 45, 50, with increasing cluster variance.

(b) Clustering result of MNIST (digit '4' forms the normal group): cluster 5, 10, 15, 20, 25, 30, 35, 40, 45, 50 with increasing cluster variance.

Figure 4: Results from Distribution Clustering.

# B    CONVENGENCE

To investigate the convergence behavior, we plotted the AUROC, AUPRC and F-score for 20 iterations for the KDDCUP experiment in Fig. 5 in the appendix, where the ground truth anomaly ratio is 20%. It demonstrates the convergence as iteration progresses. The same behavior was observed throughout our experiments.

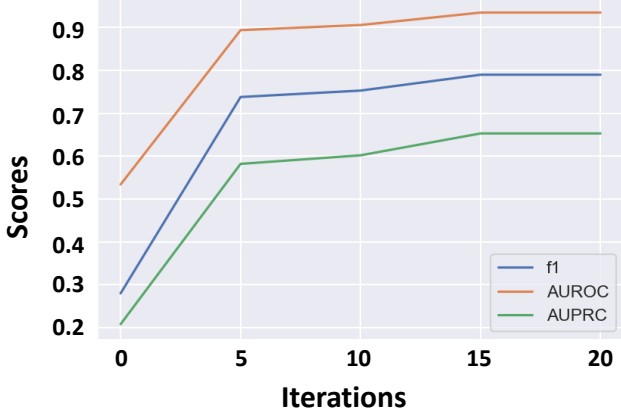

Figure 5: F1, AUROC, AUPRC scores for 20 iterations. Results obtained from experiments on KDDCUP (Lichman et al. (2013)) dataset with 20% anomaly ratio.

## C  RESULT BREAKDOWN OF UCF CRIME

Video with less than 65 segments or anomaly ratio higher than 0.35 are not taken into consideration.

Results obtained after running 5 rounds of experiments. (take average and standard deviation)

Table 7: Breakdown of Area Under the curve of the Receiver Operating Characteristic (AUROC) in % of detecting crime scene as anomalies in surveillance videos.

| Video | Crime scene | # Segment | Anomaly Ratio | AUROC (%) | Average AUROC (%) |
|-------|-------------|-----------|---------------|-----------|-------------------|
| Arson010 | Arson | 197 | 0.11 | $72.5 \pm 1.4$ | |
| Arson022 | Arson | 540 | 0.06 | $72.0 \pm 2.2$ | 67.8 |
| Arson035 | Arson | 89 | 0.21 | $60.0 \pm 5.6$ | |
| Arrest001 | Arrest | 148 | 0.13 | $75.0 \pm 1.1$ | 70.6 |
| Arrest007 | Arrest | 196 | 0.23 | $66.1 \pm 2.8$ | |
| Burglary005 | Burglary | 483 | 0.21 | $83.0 \pm 1.4$ | |
| Burglary017 | Burglary | 132 | 0.21 | $70.0 \pm 1.0$ | 79.2 |
| Burglary018 | Burglary | 70 | 0.3 | $85.2 \pm 1.4$ | |
| Burglary079 | Burglary | 928 | 0.20 | $78.7 \pm 1.2$ | |
| Fighting018 | Fighting | 86 | 0.25 | $81.0 \pm 0.4$ | |
| Fighting033 | Fighting | 69 | 0.25 | $86.9 \pm 5.2$ | 77.1 |
| Fighting042 | Fighting | 139 | 0.28 | $63.4 \pm 2.4$ | |

## D  NETWORK PARAMETERS

Table 8 reports the settings and parameters for each experiments.

Note: for video with small number of segments, the value of p is set to be larger because anomalous samples take larger proportion given the small number of total segments.

Note: Relu is adopted as the activation function except for static data where tanh is used instead.

## E  PARAMETER OF DISTRIBUTION CLUSTERING

Please refer to the original work on distribution clusteringLin et al. (2018) for more information on parameters. Table 9 reports the parameters used for distribution clustering.

## F  SAMPLE CATVSDOG DATA

Figure 7 and 6 presents some example images of dogs and cats from ASIRRA dataset.

Table 8: Parameters used for experiments.

| Dataset | Encoder layers | # Epoch | Mini-batch size | learning rate | p (%) | r |
|---|---|---|---|---|---|---|
| KDDCup | [60, 30, 10] | 200 | 1024 | 0.001 | 30 | 10 |
| Mnist | [1028, 512, 128, 60, 10] | 500 | 1024 | 0.0005 | 20 | 5 |
| Mnist (raw pixel) | [512, 256, 128, 60, 10] | 300 | 128 | 0.001 | 10 | 10 |
| Cifar-10 | [1028, 512, 128, 60, 10] | 500 | 1024 | 0.001 | 10 | 5 |
| CatVsDog | [1028, 512, 128, 60, 30] | 500 | 500 | 0.001 | 25 | 5 |
| Arson010 | [1028, 512, 128, 60, 10] | 200 | 128 | 0.0001 | 30 | 10 |
| Arson022 | [1028, 512, 128, 60, 10] | 200 | 128 | 0.0001 | 20 | 5 |
| Arson035 | [1028, 512, 128, 60, 10] | 200 | 64 | 0.0001 | 30 | 10 |
| Arrest001 | [1028, 512, 128, 60, 10] | 200 | 128 | 0.0001 | 30 | 10 |
| Arrest007 | [1028, 512, 128, 60, 10] | 250 | 128 | 0.0001 | 30 | 10 |
| Burglary005 | [1028, 512, 128, 60, 10] | 200 | 200 | 0.0001 | 20 | 5 |
| Burglary017 | [1028, 512, 128, 60, 10] | 200 | 64 | 0.0001 | 30 | 10 |
| Burglary018 | [1028, 512, 128, 60, 10] | 200 | 32 | 0.0001 | 30 | 10 |
| Burglary079 | [1028, 512, 128, 60, 10] | 200 | 128 | 0.0001 | 20 | 5 |
| Fighting018 | [1028, 512, 128, 60, 10] | 200 | 64 | 0.0001 | 30 | 10 |
| Fighting033 | [1028, 512, 128, 60, 10] | 200 | 32 | 0.0001 | 30 | 10 |
| Fighting042 | [1028, 512, 128, 60, 10] | 200 | 64 | 0.0001 | 30 | 10 |

Table 9: Parameters used for distribution clustering.

| Dataset | thres | min_clus | max_dist |
|---|---|---|---|
| MNIST | 0.1 | 7 | 1.4 |
| Cifar-10 | 0.06 | 5 | 1.4 |
| CatVsDog | 0.06 | 5 | 1.4 |
| video experiments | 0.05 | 4 | 1.5 |

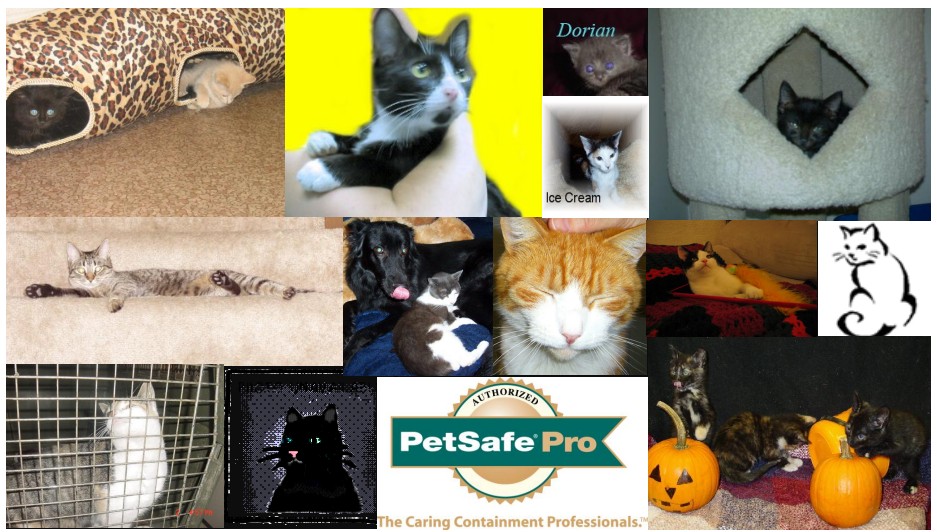

Figure 6: example cat images from ASIRRA.

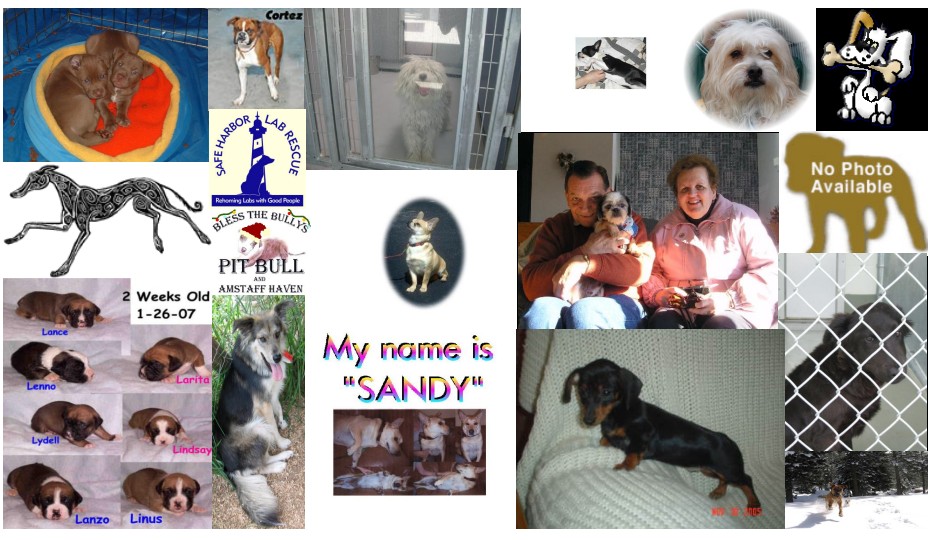

Figure 7: example dog images from ASIRRA.

