# OpenReview forum: "Deep End-to-end Unsupervised Anomaly Detection "
_ICLR.cc/2020/Conference — Reject_

### Official Review · AnonReviewer3 · 2019-10-21
**Official Blind Review #3**

**Rating:** 3

**Review:**

In the article, the authors solve the problem of anomaly detection using a fully unsupervised approach. They try to deal with the main challenge of anomaly detection: a lack of certainty on what defines normal data and anomaly one. For this purpose, the authors iteratively use: 1) autoencoders to learn the representation of the data; 2) applying in the latent space clustering to get a new training set and retrain autoencoders. The experimental results show that the author’s method performed better results than such a baseline model as one-class SVM and one-class NN.

The proposed algorithm looks robust and well-motivated, but the text of the article and the experiments can be improved. As the proposed approach is a heuristic, the experiments should be done more persuasively, including more metrics used and more alternative algorithms considered.

The key comments are the following:
1. The formatting of the article needs to be improved e.g.:
2. there is no comma between rows in the equation (1) ;
<<to be accepted into the “training” set, .>> - there is an extra comma;
3. round brackets in the equation (6) should be bigger;
4. Table 3 is bigger than the page sizes.
5. The quality of the pictures should be improved:
- Increase the captions font size in Figure 2;
- The captions and the legends in Figure 3 are practically not visible;
6.  Is the DAGMM method SOTA in anomaly detection with deep autoencoder? There are many other methods with similar ideas. We expect that we should provide a comparison with other methods:
https://arxiv.org/pdf/1809.02728.pdf - IGMM-GAN
https://papers.nips.cc/paper/7915-generative-probabilistic-novelty-detection-with-adversarial-autoencoders.pdf - GPND AE
6. Also, DAGMM works badly according to the experiments in the article with max AUROC in Table 1 only 50.3 (so it seems that it is no better than the coin-flipping)
7. Why was the only selected digit for analysis 4? Usual for comparison anomaly detection on MNIST dataset apply the following procedure: for each figure in dataset consider corresponded class as anomaly data, and the rest of the digits are used as normal data, e.g.:
https://arxiv.org/pdf/1802.06222.pdf
https://arxiv.org/pdf/1906.11632.pdf
8.  Class imbalances can affect the value of the AUROC metric. Possibly, the other metrics like AUPRC, F1-scores will better reflect the work of the algorithms for comparison. Also, AUROC is not representative when it comes to the selection of the threshold for anomaly detection. Precision and Recall can help to get more insights.
9. In Table 3, the result of applying the proposed algorithm presented with standard deviation, but other methods are represented by one metric value. Why? The explanation is required.

**Experience Assessment:**

I have published one or two papers in this area.

**Review Assessment: Checking Correctness Of Derivations And Theory:**

I carefully checked the derivations and theory.

**Review Assessment: Checking Correctness Of Experiments:**

I assessed the sensibility of the experiments.

**Review Assessment: Thoroughness In Paper Reading:**

I read the paper thoroughly.

---

> ### Author Response · Authors · 2019-11-15
> **Response to Reviewer #3**
>
> Thank you for your valuable time and comments
>
> 1. Comments on formatting and typographical errors.
>
> Thank you for your careful reading. We have revised our paper per your suggestions.
>
> 2. Is the DAGMM method SOTA in anomaly detection with deep autoencoder?
>
> To our best knowledge, it is the SOTA for UNSUPERVISED anomaly detection.  We have compared against GAN methods as recommended. However, available GAN methods, such as IGMM-GAN and GPND mentioned by the reviewer, require labels of normal data for training discriminator. Our method, despite being fully UNSUPERVISED, performed competitively.
>
> For IGMM-GAN, unfortunately we did not find publicly available code, so we cannot compare the performance.
>
>  For GPND, when rerun on MNIST dataset with ‘4’ as normal data, we found out that the source code uses training label information to search for the optimal threshold for F1/AUROC calculation, so they can report very good result. This trick gives the method unfair advantage. For 20% of anomaly, they achieved F1 score of 0.966 and AUROC score of 0.9737. For 50% of anomaly, they achieve even higher AUROC of 0.9748. This makes the GPND results dubious.
>
>  We also evaluate another semi-supervised GAN method done by Zaneti2018*. This paper, is claimed to be the improved version of “Efficient GAN-based anomaly detection(https://arxiv.org/abs/1802.06222), mentioned by the reviewer. Please note that it is still semi-supervised because it requires training with normal data only. Their code is publicly available. The re-run on KDD dataset shows that the best AUC-ROC and AUC-PRC achievable is 0.9925 and 0.9325; whereas for our method, the AU-ROC on KDD dataset is 0.935. Despite being unsupervised, our method performs competitively.
>
> *Adversarially Learned Anomaly Detection(Zaneti, ICDM2018) (https://arxiv.org/abs/1812.02288)
>
> 3. Why was the only selected digit for analysis is 4?
>
> As 2 of published works*, that  we are comparing with, only use digit 4 (as normal data) for analysis, we believe this is an established benchmark. We therefore follow the protocol specified in Zhou2017 and Chalapathy2018. We tested our method by replacing digit 4 with other digits and it achieved similar results.
>
>  We note that in some literatures, such as “Efficient GAN-based anomaly detection”(https://arxiv.org/abs/1802.06222) and “A survey on GANs for anomaly detection”(https://arxiv.org/abs/1906.11632), each figure in MNIST dataset is treated as anomaly data, while the rest of the digits are treated as normal ones. We did not adopt this setup, because it is more suitable for semi-supervised methods who are given the labels of normal classes, such that discriminators from normal data can be learned. In this setup, when there’s a subset in the test data the network never “sees”, it is deemed ‘anomaly’. We do not think it is truly unsupervised.
>
> Therefore, in our evaluation we follow the methodology of the 2 afore-mentioned UNSUPERVISED works, Zhou2017 and Chalapathy2018, for a fairer comparison on the MNIST performance.
>
> *Anomaly Detection with Robust Deep Autoencoders(Zhou, KDD2017)
> (https://www.eecs.yorku.ca/course_archive/2017-18/F/6412/reading/kdd17p665.pdf)
> *Anomaly Detection using One-Class Neural Networks(Chalapathy, KDD2018)(https://arxiv.org/pdf/1802.06360v1.pdf)
>
>
> 4. Other metrics like AUPRC, F1-scores will better reflect the work of the algorithms in comparison.
>
> Thank you for the advice. We have added evaluation based AUPRC metric in our revision. Please refer to Table 3 in the revision.

---

### Official Review · AnonReviewer1 · 2019-10-21
**Official Blind Review #1**

**Rating:** 6

**Review:**

In this paper the authors propose a framework for anomaly detection. The method is based on autoencoders and reconstruction error, but instead of training the autoencoder using all the data-points, the method iteratively uses some form of clustering to determine the points which presumably belong to the normal set, and uses them for training the autoencoder. This helps make the method robust when the portion of anomalous data-points is high.

The paper is well-written and clear and the proposed architecture is novel to my knowledge. Nevertheless, I have the following concerns about the paper. Given clarifications in the author response, I would be willing to increase the score.

- In terms of novelty, separating the dataset into normal + outliers/noise is not novel (Zhou 2017 cited in the paper). The novel part here is perhaps using variance and clustering for making the separation. However, using variance is not well motivated and it is referred to Figure 4, in which the argument is not clear. Similarly, why the reconstruction error is included in the latent representation (eq 5) is not clear.

- Given the similarity of the idea to Zhou 2017, the comparison seems important (if the code is available), and also proper discussion of such similar works is required, which currently is not presented.

- How is the performance affected by very low ratio of anomalies? This can be shown by including %2,%3 of anomalies in Table 2.

- The sensitivity of the results to the choice of hyper-parameters: p0, p, and r is not clear, and how these parameters are chosen is not discussed. It would be interesting to see how the performance is affected by different choices of the hyper-parameters.

**Experience Assessment:**

I have published one or two papers in this area.

**Review Assessment: Checking Correctness Of Derivations And Theory:**

N/A

**Review Assessment: Checking Correctness Of Experiments:**

I assessed the sensibility of the experiments.

**Review Assessment: Thoroughness In Paper Reading:**

I read the paper at least twice and used my best judgement in assessing the paper.

---

> ### Author Response · Authors · 2019-11-15
> **Response to Reviewer #1**
>
> We appreciate your constructive comments.
>
> 1. How is performance affected by very low ratio of anomalies? Results on 2%, 3% anomalies?
>
> We assessed our model on datasets with smaller anomaly ratios: 1%, 2%, 3%, 4% and 5%. Both averaged AUROC and AUPRC scores indicate consistent and robust performance.
>
> | Anomaly ratio | AUROC       | AUPRC      |
> | ------------------- | ---------------|--------------- |
> | 1                        | 98.2 +- 0.1 | 90.0 +- 1.0 |
> | 2                        | 98.1 +- 0.1 | 91.8 +- 0.2 |
> | 3                        | 98.4 +- 0.2 | 92.4 +- 0.1 |
> | 4                        | 98.2 +- 0.1 | 91.3 +- 0.9 |
> | 5                        | 99.0 +- 0.3 | 94.5 +- 0.1 |
>
>
> 2. Comparison with Zhou 2017 paper?
>
> Similar to Zhou’s paper, our method does not require any noise-free training data for model training. Both Zhou’s method and our method does not need any data label. However, Zhou’s paper use L1 or L2,1 as penalty terms, while our method combined reconstruction error and the learned low-dimensional representation (bottleneck feature from AE) in the training process. In Zhou’s paper, it has been reported that the  best F1-Score (based on threshold that gave the best trade-off between precision and recall) for MNIST dataset where “4” is treated as normal data (remaining digits are anomaly) is 64%(Figure 5). In our paper, the ROC figure reported is (70.5+/-2.1)%, which validates the performance of our method. Unfortunately, Zhou’s paper does not evaluate on datasets other than MNIST.

---

### Official Review · AnonReviewer2 · 2019-10-23
**Official Blind Review #2**

**Rating:** 1

**Review:**

The paper proposed an unsupervised anomaly detection method for the scenarios where the training data not only includes normal data but also a lot of anomaly data. The basic idea of this paper is to iteratively refine the normal data subset, selected from the whole training data set. Specifically, the paper first train an auto-encoder (AE) and determine which are the normal data samples according to the reconstruction errors. Then, using the normal data to retrain the AE again, and re-select the normal samples. Repeat the above two steps until convergence.

Overall, the novelty of this paper is in doubt. Detecting anomaly by reconstruction error of AE has been explored thoroughly, and this paper only extends it to iteratively select the normal samples. The extension seems to be very straightforward.

Also, the refining process is also problematic. It will highly depend on the initial selection, and the error will be propagated to subsequent detections. How to determine which data samples are anomalous is a key to the success of the model, but the proposed method based on the variance assumption is too intuitive and not convincing.

In addition, the experimental results on the very simple MNIST task is very poor, putting the effectiveness of the proposed model in doubt.


**Experience Assessment:**

I have read many papers in this area.

**Review Assessment: Checking Correctness Of Derivations And Theory:**

I assessed the sensibility of the derivations and theory.

**Review Assessment: Checking Correctness Of Experiments:**

I assessed the sensibility of the experiments.

**Review Assessment: Thoroughness In Paper Reading:**

I read the paper at least twice and used my best judgement in assessing the paper.

---

> ### Author Response · Authors · 2019-11-15
> **Response to Reviewer #2**
>
> We thank the reviewer for the comments. We feel there might be misunderstanding that needs to be clarified.
>
> 1. “Novelty of this paper is in doubt. .. reconstruction error of AE has been explored thoroughly. Extension seems to be straightforward. “
>
> The novelty of this paper lies in the fusion of data selection (based on clustering) and representation learning for anomaly detection under a fully unsupervised setting. By incorporating clustering to provide supervisory signals, we iterate between hypothesizing normal candidate subset and representation learning. This framework is carefully engineered to distill out anomalous data and improve the learned representation of normal data.
>
> We would like to clarify that our method did not use reconstruction error alone. Instead, it combined reconstruction error and the learned low-dimensional representation (bottleneck feature from AE) in the training process. Using reconstruction error alone is insufficient for classifying normal/anomalous samples in the training process. The reasons are two-fold:
>
> First, in the early iterations, as our algorithm adopts a tight cut-off to guess the normal candidates for auto-encoder training, some excluded but true normal samples will see large reconstruction errors. Second, during the intermediate steps, it is hard to draw a decision boundary based on the reconstruction errors (scalar values) alone. Empirically, we have tested using only the reconstruction error for all our experiments and it was confirmed to perform poorly. When combining cosine error and bottleneck feature, we re-normalized the vector after concatenation. We found such combined feature more discriminative for training set selection. The bottleneck feature, working as a low-dimensional embedding, improves the similarity measure in a clustering process. We will add this discussion to the revision.
>
> 2. Dependency on initialization
>
> We discussed this issue in the submission. Our method starts with a tight variance cut-off to ensure the initial training set is as pure as possible. This means some of the true normal samples could be discarded in this initialization. Our assumption is that given partial normal data, the autoencoder would be able to generalize well on the “unseen” normal data that was discarded, and progressively recover them as iterations go on. We also presented in Fig. 4, Appendix section that empirically, low-variance clusters correspond well to normal samples, while higher cluster variances suggest the presence of anomalous samples. In Fig. 5, we showed the convergence behavior, with initial threshold p_0 (assumed anomaly percentage) set to 30%, while ground truth is 20%.
>
> 3. Problematic refining process
>
> We would like to address that the re-clustering of data in each iteration, is not solely based on reconstruction error. Our error handling mechanism which combines reconstruction error and the learned low-dimensional representation (bottleneck feature from AE), improves the performance of method from pure clustering, as validated in Tab 3, right-most column.
>
> 4. Poor result on MNIST dataset
>
> We explained in our submission that this is due to the MNIST data being innately not high-dimensional. The NetVlad feature may have been an overkill. We re-run our experiment using raw images, and it achieved AUROC 0.824 and F1-score 0.97 (based on threshold that gave the best trade-off between precision and recall). In terms of AUROC, this result is ~30% better than state-of-the-art unsupervised method, DAGMM. Also, we would like to bring to your attention that our results on more complex dataset are significantly better compared to other UNSUPERVISED methods, and comparable with semi-supervised methods. As far as we know, we are the first paper to obtain such a good performance on unsupervised anomaly detection on video data.

---

### Author Response · Authors · 2019-11-15
**General Response to All Reviewers**

We would like to thank the reviewers for their valuable time and comments.  Our reviews are mixed, with R1=Weak Accept,  R2=Reject, and R3=Weak Reject.   We believe we can address the concerns raised by the reviewers that have led to the lower scores, especially R2 who has raised concerns on the novelty of the proposed method.  We have re-run our KDD experiments on the suggested algorithms and added the AUPRC evaluation metric. In this regard, our method demonstrates the best performance among unsupervised methods, and comparable performance to semi-supervised counterparts.  We would like to emphasize that our proposed method is an UNSUPERVISED method which did not use any label information, even in the training phase. Some methods, although claimed as unsupervised, require label information, because only normal data is used in the training. For example, in the GAN-based methods, normal data is fed into the discriminator during training. Hence, methods such as IGMM-GAN or GPND (mentioned by R3) should be classified under semi-supervised. It should be noted that while comparing with semi-supervised methods can put our method into perspective, it is UNFAIR to compare our method directly with semi-supervised methods.

Moreover, we would like to highlight that our model works with the least assumption on the data itself. Our ONLY assumption is that the anomalies are not statistically dominant in the entire dataset (and for this exact reason they are anomalies by nature). We do not make assumptions on how many classes are present in the normal data, unlike the semi-supervised methods, which require pre-isolation/identification of classes of normal data. The superiority of our method is more evident in finding undefined anomaly in video data, as demonstrated in section 4.3 of the paper.

We sincerely hope the reviewers to reconsider their scores after our rebuttal, we believe we can address all concerns in the final revision.

We have incorporated the comments in our revision (please refer to revised.pdf). The changes are highlighted in red.

---

### Comment · Area_Chair1 · 2019-11-15
**Reviewers, any comments on the author response?**

Dear Reviewers, thanks for your thoughtful input on this submission!  The authors have now responded to your comments.  Please be sure to go through their replies and revisions.  If you have additional feedback or questions, it would be great to know.  The authors still have one more day to respond/revise further.  Thanks!

---

### Decision · Program_Chairs · 2019-12-19

**Decision:**

Reject

**Comment:**

The authors propose an approach for anomaly detection in the setting where the training data includes both normal and anomalous data.  Their approach is a fairly straightforward extension of existing ideas, in which they iterate between clustering the data into normal vs. anomalous and learning an autoencoder representation of normal data that is then used to score normality of new data.  The results are promising, but the experiments are fairly limited.  The authors argue that their experimental settings follow those of prior work, but I think that for such an incremental contribution, more empirical work should be done, regardless of the limitations of particular prior work.